# Single-cell analysis of psoriasis resolution demonstrates an inflammatory fibroblast state targeted by IL-23 blockade

Luc Francis [1], Daniel McCluskey[2], Clarisse Ganier[3], Treasa Jiang [1], Xinyi Du-Harpur[3], Jeyrroy Gabriel [3], Pawan Dhami[4], Yogesh Kamra[4], Sudha Visvanathan[5], Jonathan N. Barker[1], Catherine H. Smith[1], Francesca Capon [2,6] ✉ & Satveer K. Mahil [1,6] ✉

Biologic therapies targeting the IL-23/IL-17 axis have transformed the treatment of psoriasis. However, the early mechanisms of action of these drugs remain poorly understood. Here, we perform longitudinal single-cell RNA-sequencing in affected individuals receiving IL-23 inhibitor therapy. By profiling skin at baseline, day 3 and day 14 of treatment, we demonstrate that IL-23 blockade causes marked gene expression shifts, with fibroblast and myeloid populations displaying the most extensive changes at day 3. We also identify a transient *WNT5A+/IL24+* fibroblast state, which is only detectable in lesional skin. *In-silico* and in-vitro studies indicate that signals stemming from these *WNT5A+/IL24+* fibroblasts upregulate multiple inflammatory genes in keratinocytes. Importantly, the abundance of *WNT5A+/IL24+* fibroblasts is significantly reduced after treatment. This observation is validated *in-silico*, by deconvolution of multiple transcriptomic datasets, and experimentally, by RNA in-situ hybridization. These findings demonstrate that the evolution of inflammatory fibroblast states is a key feature of resolving psoriasis skin.

Skin is an archetypal barrier tissue which protects from external insults, while orchestrating innate and adaptive responses to microbial agents[1]. Our understanding of these functions has been transformed by advances in single-cell transcriptomics, which have uncovered key differences between the cellular ecosystems of healthy and inflamed skin[2–5]. Profiling of skin lesions, however, provides only a snapshot of end-stage disease, with limited insight into the sequence of events leading to the onset and resolution of pathology. In this context, a longitudinal analysis of resolving disease can illuminate early mechanisms of drug action, identifying the fundamental events that promote a return to homeostasis. While this potential is being realized in oncology (e.g., by studying evolving cell states during immunotherapy)[6,7], inflammatory conditions remain relatively underexplored[8,9].

Psoriasis is a chronic, immune-mediated skin disorder affecting up to 2% of individuals of European descent. It presents with red, disfiguring skin plaques and is associated with multi-morbidity, poor quality of life and increased cardiovascular mortality[10]. Excessive activation of the IL-23/IL-17 axis is a key disease driver. IL-23 produced by dendritic cells and macrophages promotes the differentiation of T17 cells, leading to the release of IL-17. This cytokine, which acts in synergy with TNF, causes keratinocytes to proliferate and produce chemokines that attract further T17 cells to sites of inflammation. The ensuing feedback loop sustains chronic inflammation and epidermal hyperplasia, two hallmarks of psoriasis[11].

The IL-23/IL-17 immune axis has been successfully targeted by powerful biologic therapies inhibiting IL-17 or IL-23. IL-23 antagonists

[1]St John's Institute of Dermatology, King's College London and Guy's and St Thomas' NHS Foundation Trust, London, UK. [2]Department of Medical and Molecular Genetics, King's College London, London, UK. [3]Center of Gene Therapy and Regenerative Medicine, King's College London, London, UK. [4]Genomics Research Platform, King's College London NIHR Biomedical Research Centre, London, UK. [5]Boehringer Ingelheim Pharmaceuticals, Ridgefield, USA. [6]These authors contributed equally: Francesca Capon, Satveer K. Mahil. ✉e-mail: francesca.capon@kcl.ac.uk; satveer.mahil@kcl.ac.uk

have proven particularly efficacious, with rates of skin clearance (remission) exceeding 60% after 1 year[12]. However, the early events leading to drug-induced remission are poorly understood. Skin biopsies have been profiled during trials of IL-17 and IL-23 antagonists[13,14], but these studies have been limited by the poor resolution of bulk RNA-sequencing and the use of samples obtained weeks after the onset of clinical benefit.

Here, we seek to elucidate the early events underlying the effects of IL-23 blockade, with a view to uncovering key processes regulating skin homeostasis. We perform longitudinal single-cell RNA-sequencing (scRNA-seq) of psoriasis skin, in individuals receiving the IL-23 inhibitor risankizumab. We generate a high-resolution atlas of early disease resolution, identifying a pro-inflammatory *WNT5A+/IL24+* fibroblast state that is enriched in psoriasis lesions. This population, which shows a marked upregulation of TNF and IL-17 signaling, starts to decline in abundance 3 days after treatment initiation. Importantly, an early reduction in *WNT5A+/IL24+* fibroblasts is also detectable in patients receiving other systemic and topical therapeutics, suggesting that the normalization of this cell state is critical to the resolution of psoriatic inflammation.

## Results

### Single-cell RNA sequencing identifies key cell populations responding to IL-23 blockade

To investigate the early effects of IL-23 inhibition, we obtained serial, full-thickness skin biopsies from five individuals with psoriasis who had achieved skin clearance (remission) by 16 weeks of risankizumab treatment (Supplementary Table 1, Fig. 1A, B). We sampled lesional skin at baseline (day 0, when we also obtained non-lesional biopsies) and at two early time points (day 3 and day 14 of treatment) before full resolution of inflammation was clinically apparent. Following scRNA-seq, we generated expression profiles from 164,553 viable cells (Supplementary Table 2). We analyzed this dataset with Seurat[15], identifying cell clusters corresponding to the main cell populations found in the epidermis (keratinocytes and melanocytes), dermis (fibroblasts, vascular and lymphatic endothelial cells, pericytes, epithelial cells) and in the psoriasis immune infiltrate (mast cells, myeloid cells and T cells) (Fig. 1C, D; Supplementary Fig. 1).

Differential expression analysis (day 3 vs baseline) showed that the majority of day 3 changes were observed in myeloid cells (25 differential expressed genes, DEG) and fibroblasts (25 DEG), identifying these populations as early responders to IL-23 inhibition (Fig. 1E). Ligand-receptor analyses undertaken with CellChat[16] demonstrated a widespread reduction in the number of cell-cell interactions after 3 days of treatment (Fig. 1F). In keeping with the results of the differential expression analysis, the number of interactions originating from fibroblasts and myeloid cells was markedly reduced (Fig. 1F).

By day 14, myeloid cells and keratinocytes were displaying the strongest gene expression changes (121 and 115 DEG, respectively) (Fig. 1E). The ligand-receptor analyses were broadly consistent with these observations, showing that the number of interactions driven by myeloid cells was further reduced after 14 days of treatment (Fig. 1F).

Thus, myeloid cells and fibroblasts are the cell types showing the earliest response to IL-23 inhibition, with keratinocyte changes occurring shortly after. Importantly, the day 3 cellular shifts were observed before treatment effects became clinically apparent. This mirrors previous observations of gene expression changes preceding clinical remission (weeks 2–4 of treatment), reported in psoriasis patients receiving other biologics[17,18].

### IL-23 blockade affects cell proliferation and IL-17 signaling in keratinocytes

To investigate the effects of IL-23 inhibition in further detail, we sought to define population subsets for each cluster. In the epidermis, melanocytes could not be further subdivided, but keratinocytes could be

classified in five distinct sub-clusters (Fig. 2A and Supplementary Fig. 2A, B). These corresponded to basal (*KRT15+*), follicular (*KRT17+*), proliferating (*MKI67+*), spinous (*KRT1+/SPRR2E-*) and supra-spinous (*KRT1+/SPRR2E+*) populations (Fig. 2B). In keeping with the early clinical improvement of skin lesions (Fig. 1B), the abundance of proliferating keratinocytes was significantly reduced by day 14 (Fig. 2C).

The largest number of DEG was observed in supra-spinous keratinocytes (*n* = 228 at day 14 vs day 0) and spinous keratinocytes (*n* = 193 at day 14 vs day 0) (Fig. 2D). The downregulated genes detected at day 14 in these populations included inflammatory molecules (e.g., *IL36G, S100A7/A8/A9*) and structural proteins (*GJB2, KRT16/17*) (Supplementary Fig. 2C), known to be over-expressed in the upper layers of the psoriasis epidermis[5,19]. Conversely, the analysis of basal keratinocytes showed the upregulation of *CCL27* and *KRT15*, two genes that are lowly expressed in lesional skin[5,19] (Supplementary Fig. 2C).

Pathway enrichment analysis of the DEG identified a significant over-representation of IL-17 related genes ("Role of IL-17A in psoriasis"; FDR < $10^{-4}$ in supra-spinous keratinocytes, FDR < $10^{-2}$ in spinous keratinocytes), confirming the suppressive effect of IL-23 inhibition on T17 activation. Likewise, an upstream regulator analysis demonstrated a significant enrichment for genes that are induced by IL-17 and its synergistic partner TNF ($z < -2.50$ for both in supra-spinous keratinocytes, FDR < $10^{-5}$). Conversely, the impact of risankizumab on IFN-γ (another inflammatory cytokine induced downstream of IL-23) was limited, as the activation score ($z = -1.97$) did not reach statistical significance ($z < -2.0$).

Thus, the main effect of IL-23 blockade in the epidermis is a marked downregulation of IL-17 signaling in spinous and supra-spinous keratinocytes.

### IL-23 inhibition downregulates cytotoxic T cell activity and increases dendritic cells (DC)−3 signaling

Our initial clustering identified three immune cell types: mast cells, T cells and dendritic cells (DC). While mast cells could be subdivided into three sub-clusters (*TPSAB1-/IL1RL1+, TPSAB1+/IL1RL1+, TPSAB1+/IL1RL1-*) (Supplementary Fig. 3A, B), the frequencies of these populations did not change with treatment (Supplementary Fig. 3C). The number of DEG observed in each cluster was also low (<15 at day 14), suggesting that IL-23 inhibition has minimal effects on mast-cell activity.

T cells formed six distinct populations, including T helper (*CD4OLG+*), Treg (*CTLA4+/FOXP3+*), proliferating Treg (*MKI67+/CTLA4+/FOXP3+*), terminally differentiated effector memory (TEMRA) T cells (*CD8+/CCL5+*), cytotoxic T17 cells (*CD8+/IL17A+*) and activated *PLCG2+* cells. A *CD3-/XCL1+* cluster was also identified, corresponding to natural killer/innate lymphoid cells (NK/ILC) (Fig. 3A, B, and Supplementary Fig. 3D, E). Differential expression analysis identified TEMRA T cells as the population showing the strongest response to IL-23 blockade, with 28 DEG detected at day 3 and 30 at day 14 (Fig. 3C). These included several genes related to effector function (e.g., *GZMA* and *GZMB* at day 3; *NKG7, KLRB1* and *IL7R* at day 14), suggesting a reduction in cytotoxic activity (Fig. 3D).

The analysis of the myeloid compartment identified 10 *LYZ+* sub-clusters. These broadly recapitulated the main mononuclear phagocyte populations described by Reynolds et al. in psoriatic skin[5]. In fact, we observed cDC1 (*CLEC9A+*) and proliferating cDC1 (*CLEC9A+/MKI67+*) cells, two cDC2 populations (*CLEC10A+/CD1C+*), two DC3 populations (*IL1B+/IL23A+*), mregDC (*LAMP3+/BIRC3+*), Langerhans cells (*CLEC4A+/CD207+*) and two macrophage populations (*CD68+/CD163+*). A LYZ- sub-cluster was also detected and identified as a plasmacytoid DC (pDC) population, based on *IRF7* expression (Fig. 3E, F, and Supplementary Fig. 3F, G). Of note, the expression profile of myeloid subsets was in keeping with that described by Nakamizo et al. in their single-cell dissection of psoriasis skin[20]. mregDC were the main source of *IL15* and *IL32*, while the two

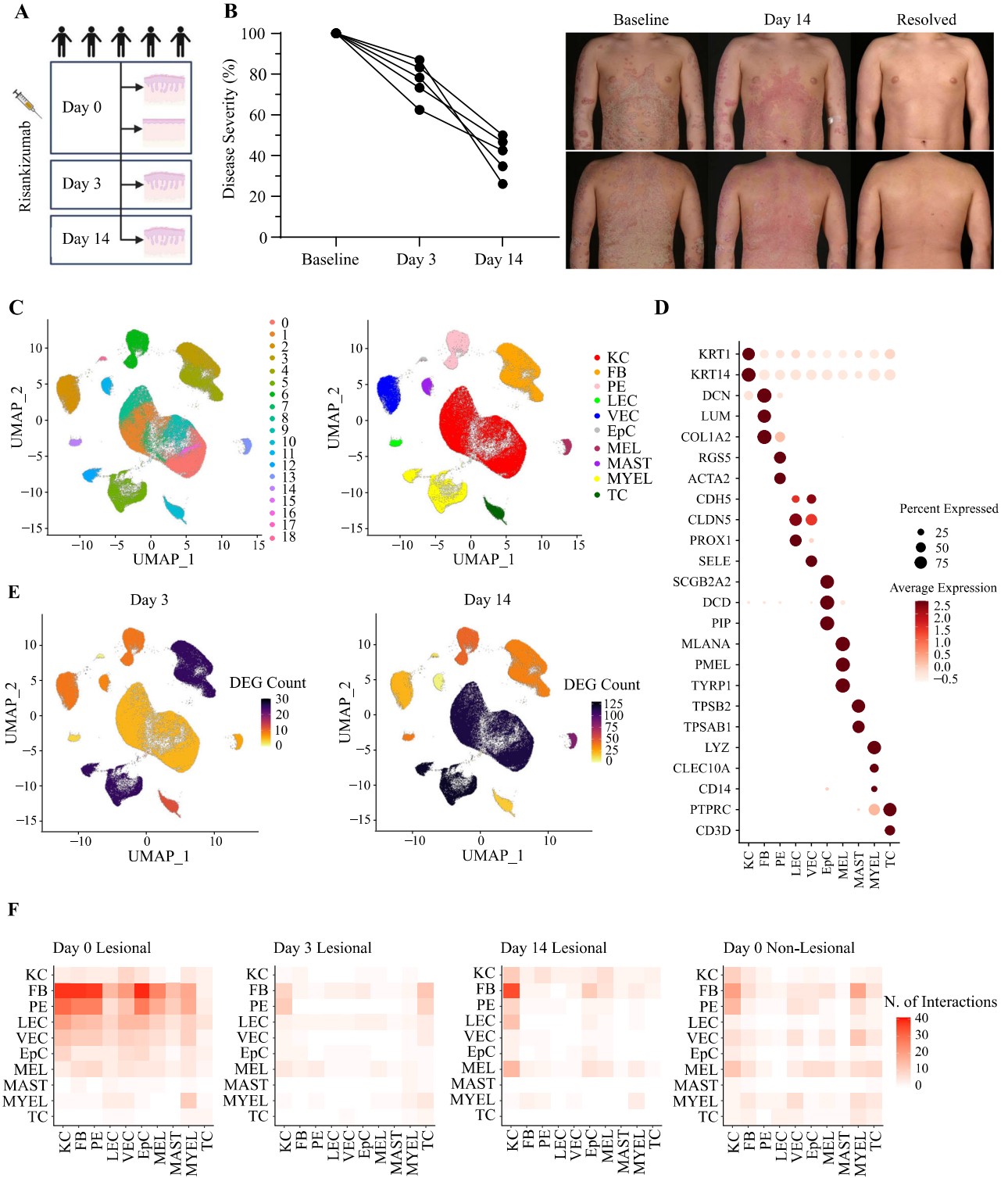

**Fig. 1 | Identification of cell populations showing an early response to IL-23 blockade. A** Experimental design of the study, showing timepoints for the sampling of non-lesional (day 0) and lesional (day 0, day 3, day 14) skin biopsies (*n* = 5 patients). Created with BioRender.com. **B** Left: Percentage reduction of disease severity (measured by the psoriasis area and severity index, PASI) after 3 and 14 days of risankizumab treatment. Every line represents a patient. Right: representative skin images showing clinical improvement during risankizumab treatment. **C** Unifold Manifold Approximation and Projection (UMAP) of 164,553 single cells (lesional and non-lesional skin), visualized as clusters (left) or skin cell types (right). **D** Dot plot showing the expression of marker genes used for the annotation of cell identities. **E** UMAP visualization of lesional skin showing the number of differentially expressed genes (DEG) observed in each cluster, after 3 (left) and 14 (right) days of treatment. **F** Heatmap depicting the number of ligand-receptor interactions between cell types. Each ligand-receptor interaction was assigned to the condition in which it had the strongest score. Interactions were then counted. The cell types expressing the ligand are labeled on the y-axis, and those expressing the receptor are labelled on the x-axis. KC, keratinocytes; FB, fibroblasts; PE, pericytes; LEC, lymphatic endothelial cells; VEC, vascular endothelial cells; EpC, epithelial cells; MEL, melanocytes; MAST, mast cells; MYEL, myeloid cells; TC, T cells.

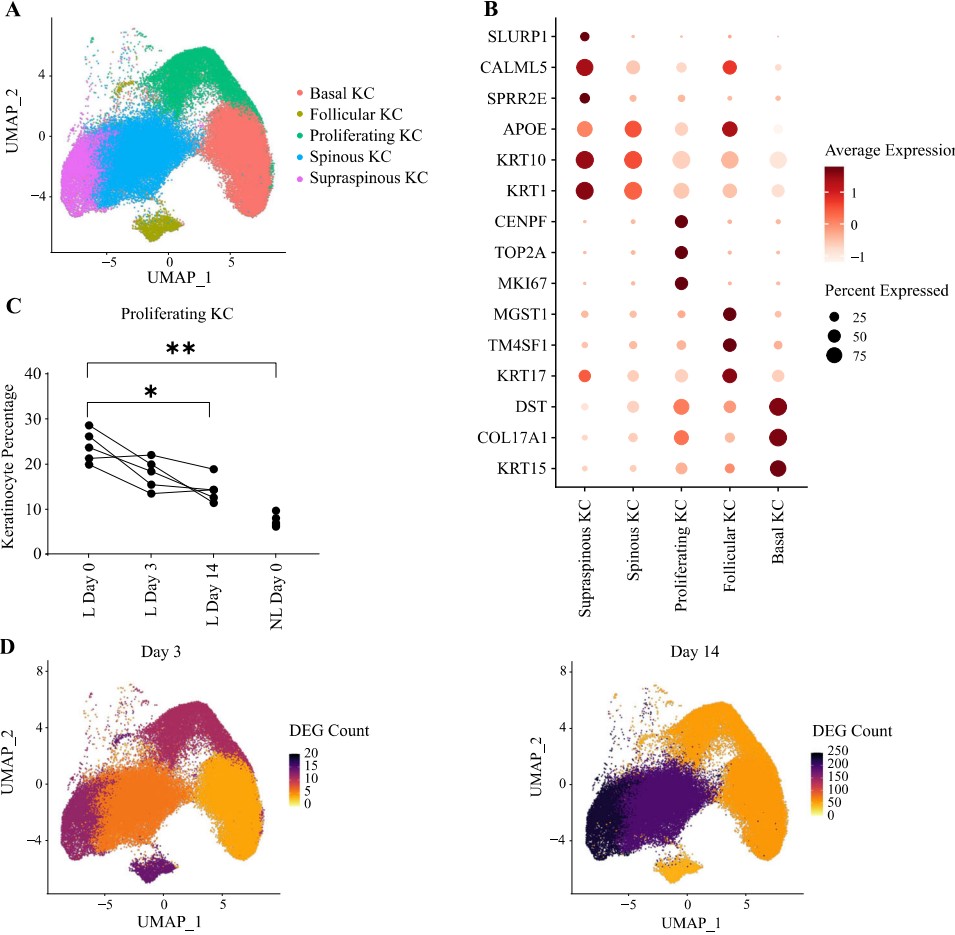

**Fig. 2 | Effects of IL-23 inhibition in keratinocytes. A** UMAP of 68,695 keratinocytes forming five distinct subclusters (*n* = 5 patients). **B** Dot plot showing the expression of marker genes used for the annotation of cell identities. **C** Abundance of proliferating cells (expressed as keratinocyte percentage) at different time points after risankizumab treatment. Every line represents a patient. *$P < 0.05$,

\*\*$P < 0.01$ (repeated measures one-way ANOVA with Dunnett's post-test). **D** UMAP visualization of keratinocytes from lesional skin, showing the number of differentially expressed genes (DEG) observed in each cluster, after 3 (left) and 14 (right) days of treatment. KC, keratinocytes; L, lesional skin; NL, non-lesional skin.

DC3 subsets expressed *IL23A, IL1B, CXCL8, IL10* and the glucose transporter *SLC2A3*. We were also able to reproduce the expansion of DC3, mregDC and cDC1 cells in lesional skin. Our analysis showed that the abundance of these cell subsets is not affected during the early stages of IL-23 blockade (Supplementary Fig. 3H). Conversely, day 14 vs day 0 differential expression analysis identified the two DC3 subsets as the cells showing the strongest response to risankizumab (180 DEG detected in DC3-2 and 155 in DC3-1) (Fig. 3G). DEG included *IL23A, CXCL3* and *IL1B*, all of which were upregulated at day 14. Notably, the expression of *TNFAIP3* (a negative regulator of IL-17 and NF-κB signaling) was also increased (Supplementary Fig. 3I), suggesting that the elevation in cytokine gene expression is a transient phenomenon, which could be caused by the modulation of negative feedback loops regulating IL-17/IL-23 signaling[21].

### IL-23 blockade downregulates mediators of leukocyte diapedesis in pericytes and endothelial cells

While fibroblasts (described below) were the most abundant cell type in the dermis (Supplementary Fig 1B), we also observed epithelial cells, lymphatic endothelial cells, vascular endothelial cells and pericytes. Neither epithelial cells nor lymphatic endothelial cells could be further subdivided, while vascular endothelial cells formed five sub-clusters (Fig. 4A, Supplementary Fig. 4A, B). These included arterioles (*HEY1+/S100A4+*), capillaries (*PLVAP+/RGCC+; ACKR1+/FABP4+*), post-capillary venules (*ACKR1+/SELE+*) and venules (*ICAM1+/VWF+*) (Fig. 4B).

Although the number of DEG observed at day 14 was <40 in all populations, there was evidence for a significant downregulation of IL-17 signaling in lymphatic endothelial cells and *ACKR1+/FABP4+* capillaries FDR < $10^{-7}$ in both). Downregulation of genes promoting leukocyte chemotaxis (*S100A7, S100A8, S100A9*) and extravasation (*COL4A1, COL4A2*) was also observed in *PLVAP+/RGCC+* capillaries (Fig. 4C).

Pericytes formed two clusters (pericyte 1 and 2), distinguishable based on *PDGFRB* and *MYL9* expression (Fig. 4D, E, and Supplementary Fig. 4C, D). Of note, pericyte 1 expressed *IL17A*, similarly to the *PDGRFB+* pericyte-like fibroblasts described by Gao et al.[22]. Pericyte 1 cells also showed the largest number of DEG (*n* = 63, day 14 vs day 0) (Fig. 4F), including several genes implicated in leukocyte chemotaxis (*CXCL8, S100A7, S100A8, S100A9*) and diapedesis (*VIM, ACTA2, SPARC*) (Fig. 4G).

Thus, IL-23 blockade causes an early downregulation of the intermediate filaments, extracellular matrix proteins and chemokines that facilitate the migration of leukocytes towards sites of skin inflammation.

### IL-23 inhibition reduces the abundance of a pro-inflammatory, *WNT5A+/IL24+* fibroblast population

Fibroblasts formed 11 distinct sub-clusters (Fig. 5A and Supplementary Fig. 5A, B). These included *CCN5+/PI16+, COL11A1+/DPEP1+* and *ANGPTL7+* populations, as well as three *COMP+* sub-clusters, three

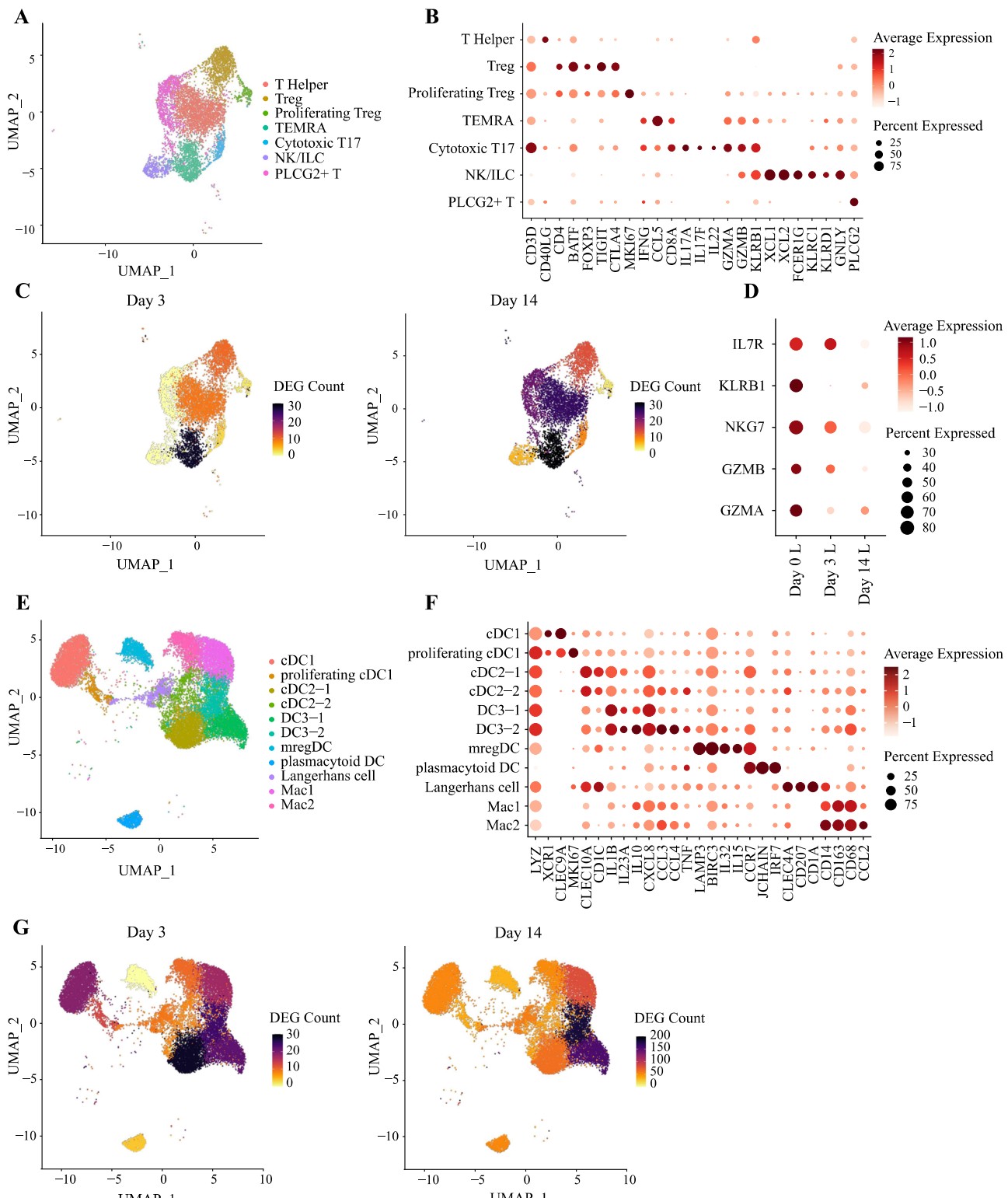

**Fig. 3 | Effects of IL-23 inhibition on lymphocytes and dendritic cells. A** UMAP of 6263 lymphocytes forming seven distinct clusters (*n* = 5 patients). **B** Dot plot showing the expression of marker genes used for the annotation of T cell identities. **C** UMAP visualization of lymphocytes from lesional skin, showing the number of differentially expressed genes (DEG) observed in each cluster, after 3 (left) and 14 (right) days of treatment. **D** Downregulation of cytotoxicity markers in TEMRA T cells, following treatment with risankizumab. **E** UMAP of 18,544 myeloid cells,

forming 11 distinct clusters. **F** Dot plot showing the expression of marker genes used for the annotation of myeloid cell identities. **G** UMAP visualization of myeloid cells from lesional skin, showing the number of differentially expressed genes (DEG) observed in each cluster, after 3 (left) and 14 (right) days of treatment. Treg, regulatory T cell; TEMRA, terminally differentiated effector memory T cell; NK, natural killer cell; ILC, innate lymphoid cell; DC, dendritic cell; Mac, macrophage; L, lesional skin.

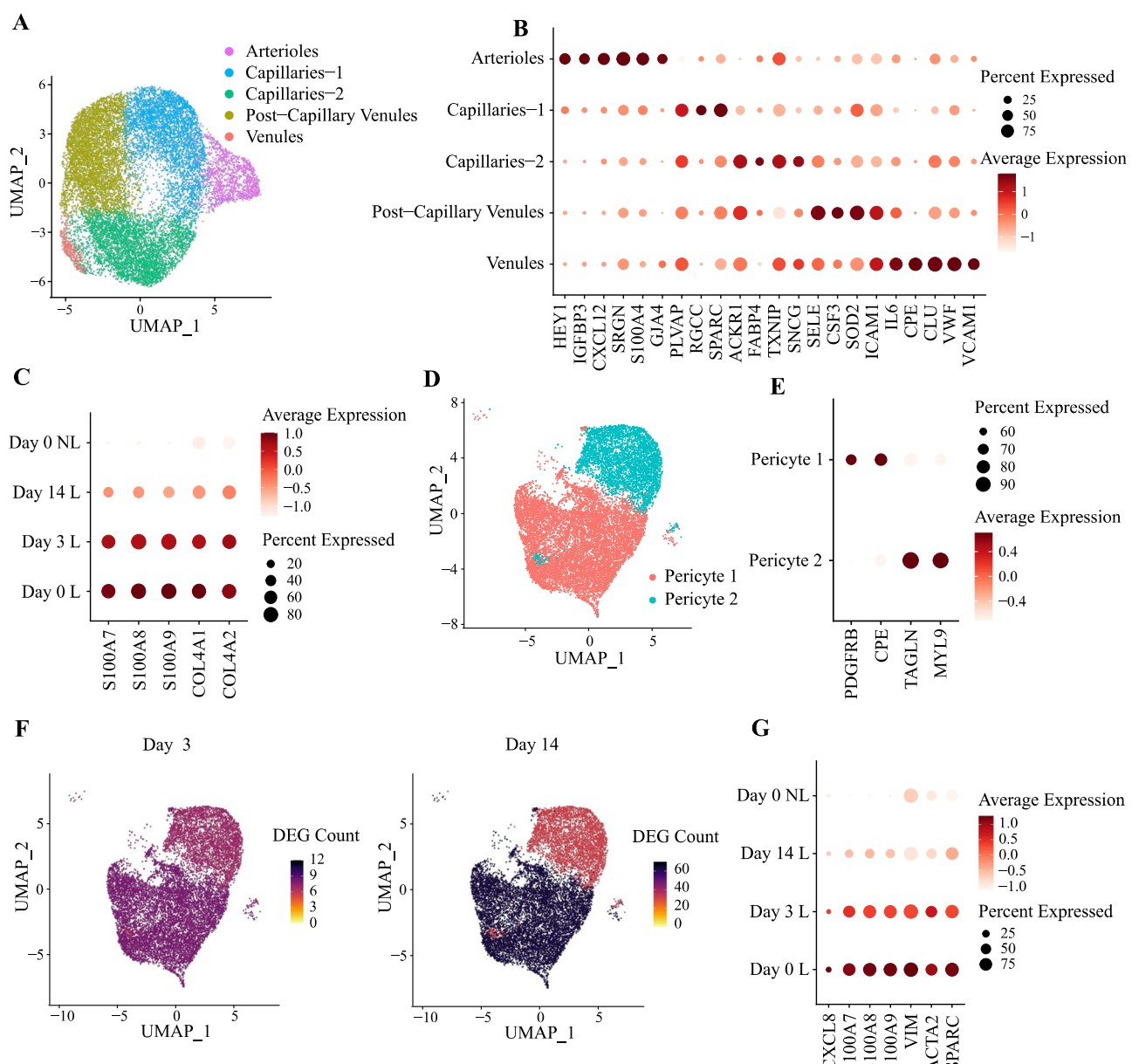

**Fig. 4 | Effects of IL-23 inhibition on vascular endothelial cells and pericytes.**
**A** UMAP of 16,420 vascular endothelial cells forming five clusters (*n* = 5 patients).
**B** Dot plot showing the expression of marker genes used for the annotation of vascular endothelial cell identities. **C** Risankizumab-induced downregulation of proinflammatory molecules and collagen genes, in *PLVAP+/RGCC+* capillaries.
**D** UMAP of 13,507 pericytes forming two clusters. **E** Dot plot showing the

expression of marker genes used for the annotation of pericyte cell identities.
**F** UMAP visualization of pericytes from lesional skin, showing the number of differentially expressed genes (DEG) observed in each cluster, after 3 (left) and 14 (right) days of treatment. **G** Genes promoting leukocyte extravasation are downregulated in pericyte 1, following risankizumab treatment. L, lesional skin; NL, non-lesional skin.

---

*APOE+* populations and a *WNT5A+/IL24+* sub-cluster (Fig. 5B). While *CCN5+/PI16+* cells showed a strong response to treatment at day 3 (143 DEG), the number of differentially expressed genes at day 14 was lower (<60) in all subclusters (Supplementary Fig. 5C). In keeping with this observation, we were able to reproduce the previously reported upregulation of *CCL19* in psoriasis fibroblasts[5]. However, we noted that *CCL19* expression was not significantly reduced by IL-23 blockade, suggesting that the modulation of this chemokine is not essential for early-stage remission (Supplementary Fig. 5D). Conversely, we found that the treatment had a marked impact on cell abundance, causing a significant decrease in the frequency of *WNT5A+/IL24+* cells. This phenomenon mirrors the absence of *WNT5A+/IL24+* cells from non-lesional skin (Fig. 5C), identifying the sub-cluster as a lesional cell state. Accordingly, an analysis of the 237

genes that were differentially expressed in *WNT5A+/IL24+* cells compared to all other fibroblast clusters demonstrated a marked enrichment for the TNF and IL-17 signaling pathways (FDR < $10^{-10}$ and $10^{-4}$, respectively). This was reflected by the expression of *TNFRSF1A, TNFRSF1B, IL17RA* and *IL17RC* (but not *IL23R*) in *WNT5A+/IL24+* fibroblasts (Supplementary Fig. 6A).

To further validate these findings, we stimulated primary human fibroblasts with IL-17A and TNF. We found that the treatment caused a significant upregulation of key marker genes that are preferentially expressed in the *WNT5A+/IL24+* population (*WNT5A, IL24, CXCL8* and *TBX3*) (Fig. 5D). Thus, IL-17 synergizes with TNF in shaping the expression profiles of *WNT5A+/IL24+* cells.

In keeping with the specificity of *WNT5A+/IL24+* cells, IL-24 signaling was only detectable in day 0 lesional skin, where CellChat

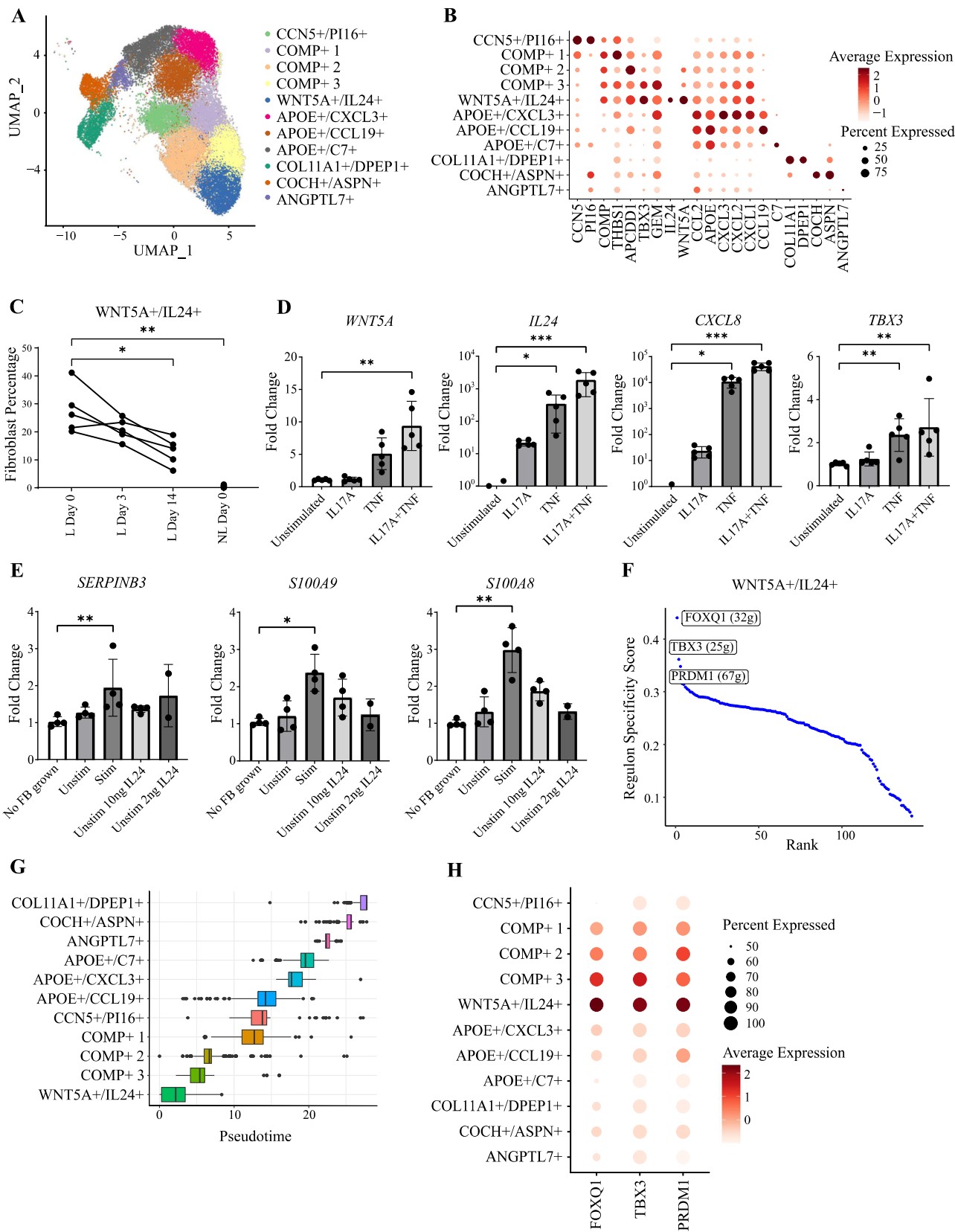

ligand-receptor analysis uncovered an interaction between the cytokine (originating from *WNT5A+/IL24+* fibroblasts) and its receptor on spinous keratinocytes (Supplementary Fig. 5E). The effects of risankizumab on IL-24 signaling were independently validated with NicheNet[23], which showed that reduced *IL24* levels at day 14 correlated with the differential expression of IL-24 target genes in spinous keratinocytes (Supplementary Fig. 5F). Of note, the DEG included several transcription factors regulating keratinocyte proliferation and differentiation (*ID1, CEBPB, STAT1*), as well as inflammatory mediators (*S100A8, S100A9, SERPINB3*) and structural proteins (*KRT15*). These observations suggest that IL-24 produced by *WNT5A+/IL24+* fibroblasts promote the activation of spinous keratinocytes.

**Fig. 5 | Effects of IL-23 inhibition on fibroblasts. A** UMAP of 31,765 fibroblasts forming eleven clusters (*n* = 5 patients). **B** Dot plot showing the expression of marker genes used for the annotation of cell identities. **C** Plot showing the decline in *WNT5A+/IL24+* cell abundance, following treatment with risankizumab. Every line represents a patient. *$P < 0.05$; **$P < 0.01$ (repeated measures ANOVA, with Dunnett's post-test). **D** Real time PCR analysis showing that key markers for *WNT5A+/IL24+* cells are upregulated in primary human fibroblasts stimulated with IL-17A and TNF. *n* = 5 biologically independent samples; data are mean (SD); *$P < 0.05$; **$P < 0.01$, ***$P < 0.001$ (Friedman test with Dunn's post-test). **E** Real time PCR analysis of inflammatory markers in human primary keratinocytes cultured with unconditioned medium (no FB grown), supernatants from IL-17A/TNF stimulated fibroblasts (Stim), control medium (supernatant from unstimulated fibroblasts, Unstim) or control medium supplemented with IL-24. *n* = 4 biologically independent samples for all conditions, with the exception of *n* = 2 biologically independent samples for Unstim 2 ng IL-24; data are mean (SD); *$P < 0.05$; **$P < 0.01$ (Kruskal Wallis test with Dunn's post-test). **F** Plot showing the SCENIC regulons detected in *WNT5A+/IL24+* fibroblasts. The three regulons with the highest specificity scores are indicated by labels, with their size (number of regulated genes) in brackets. **G** Boxplots showing fibroblast clusters ordered by pseudotime; data are median (interquartile range, minimum-maximum). **H** Dot plot showing the expression of the top regulons in the various fibroblast clusters. L, lesional skin; NL, non-lesional skin; SD, standard deviation.

To further investigate these findings, we re-analyzed spatial transcriptomic profiles generated in lesional and non-lesional psoriasis skin[24]. By examining *WNT5A* and *COMP* as markers for our population of interest (*IL24* expression was undetectable), we confirmed that it was located in the upper dermis and expanded in lesional skin (Supplementary Fig. 6B, C). We then identified the genes that were preferentially expressed in *WNT5A+/COMP+* spots. We found that these included several markers of spinous and supra-spinous keratinocytes (*CALML5, KRT1, KRT10, SLURP1, SPRR2E*), as well as pro-inflammatory mediators detected in the in-silico ligand-receptor analysis (*S100A8, S100A9, SERPINB3*) (Supplementary Fig. 6D). These findings indicate that *WNT5A+/IL24+* fibroblasts are located in an inflammatory microenvironment, in close proximity to spinous/supra-spinous keratinocytes.

To recapitulate the ligand-receptor interactions inferred *in-silico*, we cultured IL-17A/TNF stimulated primary human fibroblasts as an in-vitro model for the *WNT5A+/IL24+* state. We then transferred supernatants from these cells to human primary keratinocytes. While the supernatants did not affect the expression of genes related to keratinocyte proliferation (*MKI67, STAT1*) or differentiation (*ID1, CEBPB*) (Supplementary Fig. 6E), they caused the upregulation of key inflammatory mediators uncovered by the in-silico ligand-receptor analysis (*S100A8, S100A9* and *SERPINB3*) (Fig. 5E). Of note, these changes in gene expression were not fully recapitulated by IL-24 supplementation of control medium (supernatant from unstimulated primary fibroblasts) (Fig. 5E), indicating a likely synergy with other cytokines.

To further characterize *WNT5A+/IL24+* fibroblasts, we carried out an analysis of gene regulatory networks with SCENIC[25]. This demonstrated an enrichment for FOXQ1, TBX3 and PRDM1 regulons in *WNT5A+/IL24+* cells (Fig. 5F). Further investigation of fibroblast trajectories using Monocle 3[26] identified the three *COMP+* subsets as the populations mapping closest to *WNT5A+/IL24+* fibroblasts (Fig. 5G). This is in line with the progressively decreasing expression of *WNT5A* (Fig. 5B) and FOXQ1-regulated genes in *COMP+3, COMP+2* and *COMP+1* cells (Fig. 5H) and their absence from all other clusters. Given the role of FOXQ1 in activating Wnt signaling[27], this suggests that the drug-induced reduction in *WNT5A+/IL24+* cell numbers reflects a progressive downregulation of Wnt-related genes, leading to the re-establishment of a physiological *COMP+* phenotype. Accordingly, we found that IL-23 inhibitor treatment caused a significant increase in the abundance of *COMP+* fibroblasts (Supplementary Fig. 5G).

### A *WNT5A+/IL24+* fibroblast state is also detectable in prurigo nodularis

To understand whether *WNT5A+/IL24+* fibroblasts are present in other inflammatory skin diseases beyond psoriasis, we sought to detect these cells in publicly available scRNA-seq datasets. We therefore retrieved single-cell profiles generated in healthy skin, prurigo nodularis and atopic dermatitis[28], and integrated them with our day 0 data (Supplementary Fig. 7A–C). We found that *WNT5A+/IL24+* signature genes were absent from healthy skin, weakly expressed in atopic dermatitis, but readily detectable in prurigo nodularis lesions (Supplementary Fig. 7D). Interestingly, a closer inspection of single-cell profiles identified some differences between prurigo nodularis and psoriasis lesions, with *WNT5A+/IL24+* fibroblasts expressing *TBX3* and *CXCL8* only in the latter (Supplementary Fig. 7E).

Taken together, our observations identify *WNT5A+/IL24+* fibroblasts as an IL-17/TNF dependent, inflammatory cell state, that in psoriasis, is rapidly normalized by IL-23 blockade.

### A reduction in *WNT5A+/IL24+* cell numbers is an early event in the response to psoriasis therapeutics

To validate the pathogenic involvement of *WNT5A+/IL24+* fibroblasts, we sought to detect these cells in other psoriasis datasets generated in the first two weeks of treatment. We retrieved three transcriptomic datasets from public repositories and used a deconvolution algorithm (CIBERSORTx[29]) to estimate the frequency of *WNT5A+/IL24+* fibroblasts in patient skin. We then compared the inferred cell abundance in different sample groups.

The first dataset we examined was generated in individuals with psoriasis who were receiving guselkumab, a biologic therapy that blocks the same IL-23 subunit targeted by risankizumab[30]. The analysis of these samples confirmed that *WNT5A+/IL24+* fibroblasts are only detectable in lesional psoriasis skin, where their abundance begins to decrease after one week of treatment (Fig. 6A). The second dataset was obtained in a placebo-controlled clinical trial of the IL-17 inhibitor ixekizumab[31]. Again, our analysis showed that the decline of *WNT5A+/IL24+* fibroblasts occurred early (2 weeks) after treatment. Importantly, this reduction in cell numbers was not observed among the individuals receiving placebo (Fig. 6B). Finally, we examined RNA-seq data generated in individuals receiving a topical glucocorticoid (halomethasone monohydrate 0.05% cream) for the treatment of psoriasis[32]. We observed that the abundance of *WNT5A+/IL24+* fibroblasts in lesional skin was significantly reduced after two weeks of effective treatment. Of note, this trend was reversed upon treatment withdrawal and clinical relapse (Fig. 6C).

To complement these *in-silico* observations, we carried out RNA in-situ hybridization in the skin of three newly ascertained patients, who had been successfully treated with risankizumab. We sampled lesional psoriasis skin at baseline (day 0, when non-lesional biopsies were also obtained) and at day 14 of treatment. The analysis of these samples confirmed that our population of interest was present in the upper dermis of lesional psoriasis skin, where its abundance was markedly reduced by IL-23 blockade (Fig. 6D, E).

Taken together, these observations identify the reduction of *WNT5A+/IL24+* fibroblasts as an early event mediating the resolution of skin inflammation in psoriasis, following systemic or topical treatment (Fig. 6F).

### Discussion

We present a comprehensive single-cell atlas of resolving psoriasis, which captures the dynamic cell states mediating a return to skin homeostasis. We specifically investigated the effects of risankizumab, as this highly effective drug targets the central IL-23/IL-17 pathogenic axis[12].

We readily detected a marked downregulation of IL-17 signaling in keratinocytes, fibroblasts and vascular endothelial cells sampled after

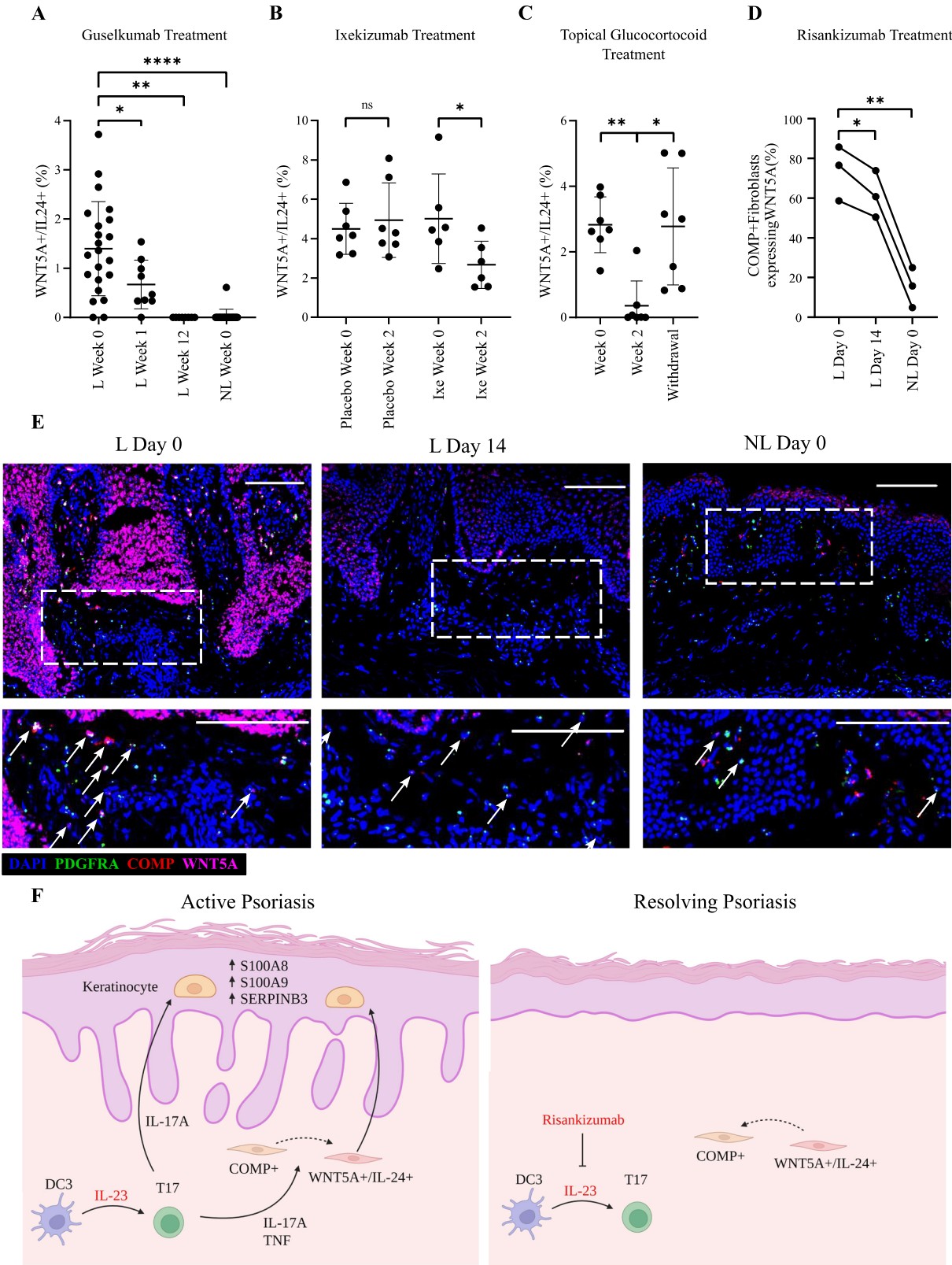

**F**  Active Psoriasis                    Resolving Psoriasis

14 days of risankizumab treatment. At this time point, the impact of IL-23 blockade on IFN-γ activity was modest, with inhibition scores only trending towards statistical significance in keratinocytes. These results complement those of a recent study where the systemic effects of IL-23 inhibition were examined by phenotyping individuals with *IL23R* deficiency. Interestingly, the analysis of this rare condition uncovered prominent defects of IFN-γ responses and only a limited impairment of

IL-17 activity[33]. Thus, the consequences of IL-23 inhibition may be organ specific. This warrants the study of IL-23 antagonists in other tissues targeted by these drugs (e.g., the intestinal mucosa[34] and the synovium[35]).

A key result from our study is the identification of a *WNT5A+/IL24+* inflammatory fibroblast state, which is expanded in lesional psoriasis skin and rapidly shrinks after treatment initiation. Interestingly, other

**Fig. 6 | The abundance of *WNT5A+/IL24+* fibroblasts declines rapidly following psoriasis treatment.** Deconvolution of transcriptome datasets shows that the frequency of *WNT5A+/IL24+* fibroblasts is significantly reduced by treatment with guselkumab (*n* = 22 patients for L Week 0, *n* = 9 for L Week 1, *n* = 8 for L Week 12, *n* = 20 for NL Week 0; data are mean (SD)) (**A**), ixekizumab (*n* = 6 for placebo group, *n* = 7 for ixekizumab group; data are mean (SD)) (**B**) or topical glucocorticoid (halomethasone monohydrate 0.05%), with the latter effect going into reverse upon drug withdrawal and clinical relapse (*n* = 7 patients per group; data are mean (SD)) (**C**). **D** The percentage of *WNT5A+* fibroblasts (*PDGFRA+* cells) detected by RNA in-situ hybridization decreases following treatment with risankizumab, (*n* = 3 patients, data are mean (SD)). **E** Representative confocal microscopy image showing the results of RNA in-situ hybridization (10x magnification, scale bar 100 μm). Arrows indicate cells triple-positive for *PDGFRA* (green), *COMP* (red) and *WNT5A* (magenta). **F** Diagram showing the early effects of therapeutic IL-23 inhibition. Left: in lesional psoriasis skin, IL-23 drives the differentiation of T17 cells. These cells produce IL-17, leading to the activation of keratinocytes and the evolution of *COMP+* fibroblasts into a *WNT5A+/IL24+* state. The ensuing IL-24 production further reinforces keratinocyte activation. Right: following IL-23 blockade, T17 cell differentiation and IL-17 production are inhibited, while *WNT5A+/IL24+* fibroblasts revert to their *COMP+* state. As a result of diminished IL-17 and IL-24 production, keratinocyte homeostasis is re-established. Created with BioRender.com. Ixe, ixekizumab; L, lesional skin; NL, non-lesional skin; SD, standard deviation; ns, not significant; *$P < 0.05$; **$P < 0.01$; ****$P < 0.0001$ (mixed effects ANOVA with Dunnett's post-test (**A**), two-sided paired *t* test (**B**), or repeated measures ANOVA with Dunnet's post-test (**C–D**)).

cell types that are altered in psoriasis (e.g., DC3, mregDC and cDC1 cells) were similarly expanded in lesional skin, but did not decrease in abundance within 2 weeks of treatment.

Our interrogation of external datasets also indicates that the decline in *WNT5A+/IL24+* cell numbers is a drug-agnostic phenomenon, suggesting that the normalization of this cell state might contribute to early, treatment induced remission.

Inflammatory fibroblasts are typified by their ability to produce and respond to cytokines. As single-cell transcriptomics have illuminated the functional diversity of stromal cells, these inflammatory subsets have been implicated in the pathogenesis of several immune mediated conditions[3,36,37]. For example, an integrated analysis of four immune-mediated disorders (rheumatoid arthritis, inflammatory bowel disease, interstitial lung disease, and Sjogren's syndrome) identified two shared pro-inflammatory states (*CXCL10+/CCL19+* and *SPARC+/COL3A1+* fibroblasts) that were also detectable in atopic dermatitis[38]. Here we observed both cell states in our samples (corresponding to *APOE+/CCL19+* and *COL11A1+* fibroblasts) and detected an upregulation of *CCL19* in lesional fibroblasts. However, neither inflammatory population showed increased abundance in lesional skin or declined in frequency following IL-23 inhibitor treatment (Supplementary Fig. 5B). A *WNT5A+/POSTN+* fibroblast subset has recently been identified in the inflammatory skin disease prurigo nodularis[39]. In affected individuals, these cells were shifted towards a cancer-associated phenotype and showed a prominent activation of type I interferon responses. Neither of these features were observed in psoriasis *WNT5A+/IL24+* cells, despite these expressing *POSTN*. Instead, our analysis indicated that a *WNT5A+/IL24+* inflammatory state can be accompanied by the upregulation of disease-specific genes such as *CXCL8* in psoriasis. As additional datasets enabling robust sub-clustering of fibroblasts become available, it will be possible to determine whether this phenomenon also applies to other IL-17/TNF mediated skin disorders. Likewise, the analysis of future datasets of treatment responses will establish if *WNT5A+/IL24+* fibroblasts undergo dynamic, treatment-dependent shifts in other inflammatory skin diseases beyond psoriasis.

Both *WNT5A* and *IL24* are overexpressed in psoriasis skin lesions[40,41]. Furthermore, the upregulation of IL-24 has specifically been detected in human dermal fibroblasts exposed to IL-17 cytokines[41], which is in keeping with our observation of enhanced IL-17 signaling in *WNT5A+/IL24+* cells. Notably, prior research has demonstrated that both Wnt5a and IL-24 promote keratinocyte proliferation and cytokine release[40,41]. Accordingly, our ligand-receptor analyses showed that *WNT5A+/IL24+* fibroblasts communicate with spinous keratinocytes in lesional skin. We also found that IL-23 inhibition abolishes this interaction, causing the downregulation of genes mediating keratinocyte activation. Our functional data supports an inflammatory crosstalk between *WNT5A+/IL24+* fibroblasts and keratinocytes, although further experiments will be required to identify the ligands mediating cell-cell communication and fully characterize their pathogenic potential. The crosstalk was further substantiated by our

in-situ hybridization experiments and re-analysis of spatial transcriptomic data, which mapped *WNT5A+/IL24+* fibroblasts to the upper dermis, in proximity to keratinocytes.

Our work also provides insight into the relationships between fibroblast populations, suggesting that the *WNT5A+/IL24+* state originates from *COMP+* cells. As dermal fibroblast plasticity has been implicated in wound healing[42], fibrosis[43] and tumor responses[44], these results add to emerging evidence that the evolution of dynamic fibroblast states is also important for the onset and resolution of chronic inflammation.

Our study has some limitations. We were unable to obtain viable neutrophils and only captured a relatively small number of T cells (~6000 in total), which prevented us from resolving rarer subsets (e.g., MAIT cells) and cell states. Similar issues have been reported in other scRNA-seq studies[45] and likely reflect cell damage caused by the skin dissociation protocol. Given our stringent inclusion criteria, our scRNA-seq discovery cohort only included male individuals of European descent without comorbidities. Thus, further investigations will be required to establish whether our findings can be generalized to other demographics.

Since the aim of this study was to investigate the early effects of IL-23 inhibition, our patient sample collection deliberately targeted the first two weeks of treatment. Nonetheless, our analysis of publicly available transcriptome data indicates that the abundance of *WNT5A+/ IL24+* cells rapidly increases upon treatment withdrawal and clinical relapse. If this finding is confirmed in future cohorts, the frequency of *WNT5A+/IL24+* cells could be monitored in the increasing number of patients receiving biologics, with the potential to inform personalized, drug tapering strategies in psoriasis remission.

## Methods

### Participants

Written informed consent was obtained from all participants, in line with approval from the London - Westminster Research Ethics Committee (REC ref 11/H0802/7). All participants were biologic-naïve, adults of European descent, with a dermatologist confirmed diagnosis of severe psoriasis (PASI > 10) (Supplementary Table 1). Self-reported sex and ethnicity were recorded. To minimize confounders, we also applied additional inclusion criteria: no comorbidities, no systemic immune-modifying therapy for at least 4 weeks prior to risankizumab treatment, no topical steroids to the sampled area for at least 5 days prior to sampling. Six to eight mm full thickness punch biopsies were obtained from participants at baseline (day 0; lesional and non-lesional skin) and after 3 and 14 days of treatment (lesional skin only). All biopsies were taken from site-matched areas on the lower back/ buttocks.

### Whole skin dissociation, scRNA-seq library preparation and scRNA-sequencing

Skin biopsies were processed immediately after collection and incubated in 5U/ml dispase (Stem Cell Technologies) at 4 °C overnight, to

separate the epidermis from the dermis. The dermis was minced and then processed using the Whole Skin Dissociation Kit (Miltenyi Biotec; 130-101-540) and a gentleMACS Tissue Dissociator (Miltenyi Biotec). The epidermis was minced and digested with a 50:50 trypsin-EDTA: Versene solution (Gibco) at 37 °C for 15 min. Dissociated cells were then filtered through a 100 µm cell strainer. Following incubation with Fc block (Biolegend), TotalSeq Hashtag antibodies (Biolegend; catalog numbers 394641, 394643, 394645, 394647; clone LNH-94, 2M2) were added to delineate lesional dermis, lesional epidermis, non-lesional dermis and non-lesional epidermis cell suspensions. Following sorting of viable cells on a FACSAria flow cytometer (BD), dermal and epidermal suspensions were mixed at a 1:1 ratio. Libraries were generated from single cells using the following kits: Chromium Next GEM Single Cell 3′ GEM, Library & Gel Bead Kit v3.1, 16 rxns (PN-1000121); Chromium Single Cell 3′ Feature Barcode Library Kit, 16 rxns (PN-1000079); Chromium Next GEM Chip G Single Cell Kit, 16 rxns (PN-1000127); Single Index Kit T Set A, 96 rxns (PN-1000213) (all from 10X Genomics). Libraries were sequenced using a NextSeq 2000 or NovaSeq 6000 instrument (Illumina) to generate 150 base pair paired end reads.

## scRNA-seq data processing

FASTQ files were aligned against the human GRCh38 reference genome using Cell Ranger v6.1.1 (10X Genomics). Cells expressing <300 or >4000 unique genes, or with greater than 20% of genes of mitochondrial origin, were removed. The scDblFinder package (v3.1.6)[46] was used to confirm that doublets had been correctly excluded. To avoid batch effects, data generated in different sequencing runs were integrated using the reciprocal principal component analysis workflow provided by the Seurat package (v4.1.0)[15]. The data were normalized and then scaled by regressing out the following variables: number of unique molecular identifiers per cell, percentage of mitochondrial genes, day of single-cell dissociation, patient identifier, and difference between G2M and S phase score.

## Cell clustering and cell type annotation

The batch-corrected coordinate space was used for linear dimensional reduction with Seurat. A K-Nearest Neighbor graph was constructed with the FindNeighbours function and unsupervised cell clustering was implemented using the FindClusters function at a resolution of 0.4). Subclustering was performed using the same approach, except the resolution was optimized separately for each population, using Clustree[47].

Genes that were differentially expressed (fold change >1.5; Bonferroni adjusted $p$ value < 0.05) in a cluster compared to all the others were identified using the FindAllMarkers function. Cell identities were then annotated by cross referencing these cluster-specific genes with published signature genes for the relevant cell type. The same approach was applied to subclusters, except small subclusters expressing multiple lineage markers were considered as doublets and removed.

## Differential expression analysis

Genes that were differentially expressed between day 0 and day 3 or day 0 and day 14 were identified with the FindMarkers function. Genes detected in <5% of cells from either sample group were removed. A negative binomial regression was applied, including patient id as a latent variable.

## Identification of enriched pathways, upstream regulators and gene regulatory networks

The Ingenuity Pathway Analysis package (Qiagen) was used to identify canonical pathways and upstream regulators that were enriched (FDR < 0.05) among the DEG identified with the FindAllMarkers and FindMarkers functions.

The gene regulatory networks that were active in lesional fibroblast subclusters at day 0 were identified using the workflow implemented by SCENIC (v1.3.0)[25]. Briefly, co-expression modules between transcription factor and target genes (regulons) were derived using GRNBoost (a scalable alternative to GENIE3), then RcisTarget was used to refine the regulons by inferring direct targets of the transcription factors. Finally, regulon activity scores were calculated for each cell, using AUCell.

## Cell-cell interaction inference

Receptor-ligand interaction analysis was performed using CellChat (v1.4.0)[16]. The main analysis was based on major cell type annotations. A separate run was performed for each sample group and the number of significant interactions was calculated for each cell type pair. For the analysis of IL-24 signaling, keratinocytes, fibroblasts, T cells, and myeloid cells were divided into subclusters. Ligand-receptor interactions originating from *WNT5A+/IL24+* fibroblasts were then queried at day 0 and day 14. The interaction between IL-24 and its receptor was also confirmed by using NicheNet (v1.1.1)[23] to analyse day 0 and day 14 lesional skin data. *WNT5A+/IL24+* cells were set as source and the various keratinocyte subclusters were in turn set as recipient cells. Results were generated using the 'nichenet_seruatobj_aggregate' function.

## Cell culture

Primary dermal fibroblasts and primary keratinocytes were isolated from healthy skin discarded after plastic surgery (REC ref 14/LO/2169). Fibroblasts were grown in Dulbecco Modified Eagle Medium (DMEM) supplemented with 10% fetal bovine serum and 1% penicillin-streptomycin (all reagents from Gibco). For the analysis of *WNT5A+/IL24+* marker genes, fibroblasts were stimulated for 24 h with 5 ng/mL TNF (R&D Systems), 50 ng/mL IL-17A (R&D Systems), 5 ng/mL TNF + 50 ng/mL IL-17A, or vehicle. Cells were then harvested for RNA isolation.

For fibroblast-keratinocyte crosstalk experiments, fibroblasts were first stimulated for 24 h with 5 ng/mL TNF + 50 ng/mL IL-17A, or vehicle. Cells were then washed with phosphate buffered saline (Gibco) and incubated in complete DMEM medium for a further 24 h. Meanwhile keratinocytes were grown in EpiLife Keratinocyte Medium supplemented with 60 µM calcium, Supplement 7 and 1% penicillin-streptomycin (all reagents from Gibco). After an overnight incubation in Supplement 7-free medium, keratinocytes were incubated for 24 h in complete Epilife Medium diluted 1:3 in either complete DMEM, culture medium from TNF/IL-17A stimulated fibroblasts, culture medium from unstimulated fibroblasts, or culture medium from unstimulated fibroblasts supplemented with IL-24 (2 ng/ml or 10 ng/ml, R&D Systems). Following RNA isolation from harvested keratinocytes, gene expression was assessed by real-time PCR using primers reported in Supplementary Table 3. Transcript levels were normalized to expression of *GAPDH* (fibroblasts) or *PKG1* (keratinocytes).

## Pseudotime analysis

Seurat fibroblast data from day 0 lesional skin were converted and imported into Monocle 3 (v1.3.1)[26] using the cell_data_set function from SeuratWrappers. Cells were separated into clusters by Leiden community detection (cluster_cells function). A cell trajectory was then fitted with the learn_graph function. Root nodes were selected within the *WNT5A+/IL24A+* population and the order_cells function was applied.

## Re-analysis of publicly available datasets

For the analysis of spatial transcriptomic profiles, the dataset generated by Castillo et al.[24] (GSE202011) was downloaded as an RDS file

 

from https://zenodo.org/records/7562864. The data were analyzed using Seurat to visualize gene expression. Spots were considered positive for the expression of genes of interest, if the relevant UMI count was ≥1. FindMarkers was then used to identify genes that were upregulated in *WNT5A+/COMP+* spots compared to all other spots.

For the analysis of *WNT5A+/IL24+* cells in healthy, prurigo nodularis and atopic dermatitis skin, the scRNA-seq datasets published by Alkon et al.[28] were retrieved from the Gene Expression Omnibus, using identifier GSE222840. The datasets were then integrated with our day 0 lesional and non-lesional scRNA-seq data. Briefly, all datasets were normalized using the SCTransform function then merged into a single Seurat object. Following principal component analysis of the normalized data, batch correction was performed using the RunHarmony function from the Harmony package[48] (v1.0.1), with sample ID and dataset variables. The main cell clusters were identified by DEG annotation after applying FindNeighbors and FindClusters. The fibroblast cluster was extracted, split by sample id and re-integrated by repeating the steps above. Finally, the fibroblasts were sub-clustered at a resolution of 0.3 and the Seurat AddModuleScore function was applied using the expression of 20 signature genes for *WNT5A+/IL24+* cells (the top DEG observed in *WNT5A+/IL24+* cells vs all other fibroblasts in baseline psoriasis skin).

For the deconvolution analysis, the scRNA-seq dataset produced in this study was down-sampled to 5000 cells and used to generate a reference signature matrix for CIBERSORTx[29]. The transcriptomic datasets generated by Sofen et al.[30], Krueger et al.[31], and Cai et al.[32], were retrieved from GEO using their respective identifiers (GSE51440, GSE31652 and GSE114729). The reference matrix was then used to infer the relative abundance of *WNT5A+/IL24+* fibroblasts.

### RNA fluorescence in situ hybridization
16μm sections were cut from skin biopsies frozen in optimal cutting temperature compound (VWR International). RNA fluorescence in situ hybridization of Hs-PDGFRA, Hs-COMP-C2 and Hs-WNT5A-C4 probes (ACD Biologics) was performed using RNAscope Multiplex Fluorescent Reagent Kit v2 (ACD Biologics; 323135) and 4-plex Ancillary kit (ACD Biologics; 232120). The manufacturer's instructions were followed with minor modifications. Fixation in 4% paraformaldehyde/PBS (Santa Cruz Biotechnology) was extended to 60 min at 4 °C, and TSA VIVID Fluorophore 520, 570 and 650 (ACD Biologics) were diluted at 1:1000. At least two representative images were captured for each section, using an Eclipse Ti-2 inverted microscope (Nikon).

### Statistical analysis
All statistical tests were implemented in GraphPad Prism (version 9.3.0 for Windows). For the analysis of scRNA-seq and RNA in situ hybridization results, the relative abundance of cell types in the four sample groups (non-lesional, lesional day 0, lesional day 3, lesional day 14) was compared using repeated measures analysis of variance (ANOVA), with a Dunnett's post-test. For the analysis of cell fractions derived by deconvolution, the relative abundance of *WNT5A+/IL24+* fibroblasts was compared across treatment groups, using a repeated measures ANOVA with a Dunnett's post-test or a paired *t* test, as appropriate. The real-time PCR data was analyzed using a Kruskal Wallis test or Friedman test, as appropriate. Both were followed by Dunn's multiple comparison test. Details of all statistical tests are reported in the relevant figure legends.

### Reporting summary
Further information on research design is available in the Nature Portfolio Reporting Summary linked to this article.

## Data availability
The scRNA-seq data generated in this study have been deposited in the NCBI Gene Expression Omnibus under accession code GSE228421. The transcriptomic datasets generated by Castillo et al.[24], Alkon et al.[28], Sofen et al.[30], Krueger et al.[31], and Cai et al.[32], used in this study are available in the NCBI Gene Expression Omnibus under accession codes GSE202011 data object can be downloaded from https://zenodo.org/records/7562864), GSE222840, GSE51440, GSE31652 and GSE114729, respectively. All other data are available in the article and its supplementary files or from the corresponding authors upon request. Source data are provided with this paper.

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

## Acknowledgements

We are grateful to all the patients who made this study possible. We also wish to acknowledge the support of St John's Institute of Dermatology Skin Therapy Research Unit (Andrew Pink, Richard Woolf, David Baudry, Isabella Tosi, John Gregory, Katherine Teather, Louise Griffiths, Qin Neville), King's College London Advanced Cytometry Platform (Isabel Correa, Richard Ellis, Leanne Farnan, Anna Rose), Nikon Imaging Centre (James Levitt, Virginia Silio, Isma Ali), and Genomics Core (Ulrich Kadolsky, Shichina Kannambath, Michelle Kleeman, Athul Menon, Rosamond Nuamah, Heli Vaikkinen). We thank Paola Di Meglio for her input on the manuscript, and Hannah Cherry, Yong-Xin Sieh and Imogen Brooks for technical assistance. This research was supported by the National Institute for Health and Care Research (NIHR) Biomedical Research Centre based at Guy's and St Thomas' NHS Foundation Trust and King's College London (guysbrc-2012-1). The views expressed are those of the author(s) and not necessarily those of the NHS, the NIHR or the Department of Health and Social Care. This work was funded by the Psoriasis Association (PhD studentship ST2/21 supporting L.F. and Grant BSTOP50/5 to C.H.S.) and the Wellcome Trust (grant 096540/Z/11/Z). DMc and T.J. were supported by the Medical Research Council (grants MR/R015643/1 and MR/W006820/1) and King's College London as a member of the MRC Doctoral Training Partnership in Biomedical Sciences. S.K.M. is funded by a NIHR Advanced Fellowship (NIHR302258). C.H.S. is supported by a NIHR Senior Investigator Award.

## Author contributions

F.C. and S.K.M conceived the study, designed the experiments with input from L.F. and obtained the necessary funding. Further funding was provided by S.V.; S.K.M. supervised the patient recruitment, which was also facilitated by C.H.S and J.N.B; Y.K. prepared the single-cell libraries under P.D.'s supervision. L.F. carried out the remaining experimental work with help from T.J. and input from S.K.M., C.G., J.G. and X.D.H; L.F. and D.Mc. carried out the computational analyses under F.C.'s supervision. F.C. and S.K.M. drafted the manuscript, which was then reviewed by L.F. with input from all co-authors.

## Competing interests

S.V. is a Boehringer-Ingelheim employee. J.N.B. has attended advisory boards and/or spoken at sponsored symposia and/or received research funding from: AbbVie, Almirall, Amgen, Boehringer-Ingelheim, Bristol Myers Squibb, Celgene, Janssen, Leo, Lilly, Novartis, Samsung, Sun Pharma. C.H.S. reports departmental research funding as investigator in EU-IMI consortia involving multiple industry partners (see biomap-imi.eu and hippocrates-imi.eu for details). F.C. has received grant support and consultancy fees from Boehringer Ingelheim. S.K.M. reports departmental income from Abbvie, Almirall, Eli Lilly, Leo, Novartis, Sanofi, UCB, outside the submitted work. The remaining authors declare no competing interests.
