## [Peer Review File · Nature Communications]

Single-cell analysis of psoriasis resolution demonstrates an inflammatory fibroblast state targeted by IL-23 blockadeREVIEWER COMMENTS

Reviewer #1 (Remarks to the Author):

This is well performed pilot study in five patients with psoriasis undergoing IL23 inhibitor therapy. The methodology is sound and analysis clearly presented. However, the reported results are entirely descriptive with no functional mechanistic data presented. The main claim, that WNT5A+IL24+ fibroblasts are psoriasis-specific, is not fully supported by the data and the importance of this population as a psoriasis driver vs bystander is not fully elucidated. The authors could do a better job relation their findings to the existing literature. The authors should address the following:

1) The observed WNT5A+IL24+ fibroblasts are called “psoriasis-specific.” However, the authors have not presented data on whether these fibroblasts exist in other inflammatory skin diseases, such as atopic dermatitis or hidradenitis suppurativa (the latter which demonstrates IL17 signaling). This could be done by obtaining public bulk RNA-seq datasets and using deconvolution methods. It could also be examined using RNA in situ hybridization of FFPE skin biopsies. The authors have not commented on how their WNT5A+IL24+ fibroblasts relate to other fibroblast populations as described in PMID: 37210042 (prurigo nodularis) and PMID: 33958582 (psoriasis).

2) The WNT5A+IL24+ fibroblasts are altered in psoriasis, but the literature indicates that many other cell types in psoriatic skin are also altered, including keratinocytes, endothelial cells, macrophages, T cells, and myeloid cells, all of which contribute to the inflammatory environment. Why should these cells be used as a “biomarker” over any other cell type? Are these cells perhaps bystanders with only a minor role in inflammation? It would be helpful to have functional data examining the effect of inhibition of these fibroblasts in vitro (or in mouse models in vivo), particularly on IL24 production and subsequent effects on keratinocytes.

3) The authors could do a much better job relating their results to other single cell studies of psoriasis in the literature, including but not limited to PMID: 34279540 (APCs), 33932468 (APCs and keratinocytes), and 35693818 (keratinocytes).

4) Much emphasis is made by the authors on the observation of molecular changes before clinical improvement. This is not new. Many studies in the past have observed molecular changes (by bulk RNA-seq and flow cytometry) in weeks 0 to 4 after therapy (e.g. PMID: 27667537, 30703387, 35757784, many others). Though 3 days is a bit sooner, the main concept has been known for a while. Please acknowledge the context of these prior studies and reduce claim of novelty.

Reviewer #2 (Remarks to the Author):

The manuscript by Francis L et al presents highly interesting data on early resolution of psoriasis skin lesions following anti-IL13 treatment. Identification of a distinct pro-inflammatory fibroblast cell population limited to active psoriasis lesions which rapidly decreases following treatment start is novel and offers insight into the early initiating molecular events in resolution of psoriasis inflammation. The study involves 5 clinically homogenous individuals biologically naive males with longstanding severe psoriasis. Skin biopsies, before treatment start, day 3 and day 14 are obtained from site-matched areas - areas however not specified - please provide this information.

Single cell analysis was performed according to established methodology, epidermal and dermal cells were disassociated and classified in clusters obtaining single cell RNA. The main finding was a psoriasis specific WNT5+/IL24 fibroblast population involved in IL17 signalling detected at Day 0 where ligand receptor interaction analysis pointed towards an interaction with its receptor on spinous keratinocytes. The WNT5+/IL24 fibroblast population declined already at Day 3 and was absent at Day 14 presumably leading to declining IL17 effect on keratinocytes identified as an early and important mechanisms for the resolution of psoriasis inflammation. Validation in public transcriptomic datasets showed similar effects in patients treated with other anti-psoriasis drugs (anti-IL17, and even topical therapy) supporting the idea that this is a general mechanism occurring in resolving psoriasis.

The data are convincing but the study would be much enhanced by adding up-to-date spatial architecture through comprehensive spatial transcriptomics revealing interacting cell clusters in situ.

Reviewer #3 (Remarks to the Author):

This is a paper by Francis and colleagues on sc analysis of psoriasis focusing on fibroblasts. The authors obtained skin biopsies from patients with psoriasis treated with anti-IL-23 neutralizing antibodies. The most prominent changes observed on day 3 were observed in myeloid cells and fibroblasts, on day 14 myeloid cells and keratinocytes. They found an overrepresentation of IL-17-related genes in keratinocytes and its downregulation in spinous and supraspinous keratinocytes. When analyzing immune cells, they found that mainly terminally differentiated effector memory T cells responded to IL-23 blockade. On day 14 certain DC3 subsets showed the strongest response to IL-23 blockade. Effects of IL-23 inhibition on vascular endothelial cells and pericytes were also investigated. Finally, the authors focused on the effects of IL-23 blockade on fibroblasts and describe a significant decrease in the frequency of WNT5A+/IL24+ cells. They also studied the possible interactions of such fibroblasts with keratinocyte proliferation and differentiation. The finding of WNT5A+/IL24+ fibroblasts in psoriatic skin and its regulation was validated in three other psoriasis datasets from public repositories (treatment with an other IL-23 inhibitor or IL-17 inhibitor or topical steroids).

This is a well-written elegant paper on psoriasis, deciphering the changes in immune and tissue cell subsets during treatment with IL-23 targeting neutralizing antibodies. The focus is on early responses at day 3 and 14.

What is missing in this paper is functional data. What exactly is the role of WNT5A+/IL24+ fibroblasts in psoriasis?

What are the signals that induce IL-24 in fibroblasts? IL-23, IL-17 or a combination of cytokines (TNF,...)?

Do the WNT5A+/IL24+ fibroblasts express the IL-23 receptor?

The authors should add some data from functional assays (e.g. in vitro co-cultures) on WNT5A+/IL24+ fibroblasts and keratinocytes. What is the impact of WNT5A+/IL24+ and IL24- (negative) fibroblast populations on keratinocyte proliferation, differentiation and production of secreted factors?

REVIEWER #1:

This is well performed pilot study in five patients with psoriasis undergoing IL23 inhibitor therapy. The methodology is sound and analysis clearly presented. However, the reported results are entirely descriptive with no functional mechanistic data presented. The main claim, that WNT5A+IL24+ fibroblasts are psoriasis-specific, is not fully supported by the data and the importance of this population as a psoriasis driver vs bystander is not fully elucidated. The authors could do a better job relation their findings to the existing literature. The authors should address the following:

1) The observed WNT5A+IL24+ fibroblasts are called “psoriasis-specific.” However, the authors have not presented data on whether these fibroblasts exist in other inflammatory skin diseases, such as atopic dermatitis or hidradenitis suppurativa (the latter which demonstrates IL17 signaling). This could be done by obtaining public bulk RNA-seq datasets and using deconvolution methods. It could also be examined using RNA in situ hybridization of FFPE skin biopsies.

Thank you for this important point. To understand whether WNT5A+/IL24+ fibroblasts exist in other inflammatory skin diseases, we have now interrogated two external scRNA-seq datasets. While publicly available studies for hidradenitis suppurativa did not include sufficient cell numbers for robust fibroblast sub-clustering, we have been able to retrieve single-cell profiles for healthy, prurigo nodularis and atopic dermatitis skin. By integrating these datasets with our day 0 samples, we have identified a WNT5A+/IL24+ fibroblast state in prurigo nodularis. We report this data in a new Results section (p.11) and in Supplementary Figure 7. Given these findings, we no-longer use the term ‘psoriasis-specific’ when referring to WNT5A+/IL24+ fibroblasts.

The authors have not commented on how their WNT5A+IL24+ fibroblasts relate to other fibroblast populations as described in PMID: 37210042 (prurigo nodularis) and PMID: 33958582 (psoriasis).

PMID 33958582 described seven mesenchymal cell clusters, which encompassed only one fibroblast population. Interestingly, however, the authors identified IL17 expression in PDGFRB+ pericyte-like fibroblasts. We observed the same expression pattern in lesional pericytes and now mention this in the main text (p.8).

The prurigo nodularis dataset generated in PMID 37210042 formed the basis of the re-analysis work described above. This showed the presence of a WNT5A+/IL24+ state in prurigo nodularis, but also uncovered some dissimilarities with the WNT5A+/IL24+ fibroblasts detected in psoriasis. We comment on those points of difference in the main text (p.11) and the Discussion (p.14).

2) The WNT5A+IL24+ fibroblasts are altered in psoriasis, but the literature indicates that many other cell types in psoriatic skin are also altered, including keratinocytes, endothelial cells, macrophages, T cells, and myeloid cells, all of which contribute to the inflammatory environment. Why should these cells be used as a “biomarker” over any other cell type? Are these cells perhaps bystanders with only a minor role in inflammation?

We have added data to show that other cell types that are altered in psoriasis (e.g., mregDCs, cDC1 and DC3 cells) (p.7 and Supplementary Figure 3H) do not decrease in frequency within 14 days of treatment initiation. Having said that, the reviewer’s point is well taken and we have rephrased our conclusions to state that the normalization of the WNT5A+/IL24+ state “might contribute” to early, drug-induced remission (p.13).

It would be helpful to have functional data examining the effect of inhibition of these fibroblasts in vitro (or in mouse models in vivo), particularly on IL24 production and subsequent effects on keratinocytes.

We generated an in-vitro model of the *WNT5A+/IL24+* state by treating primary human fibroblasts with IL-17A/TNF. We found that removal of TNF from the stimulation mix markedly reduced the expression of *IL24* (Figure 5D).

Of note, transfer of supernatants from IL-17A/TNF stimulated fibroblasts to primary human keratinocytes led to the upregulation of pro-inflammatory mediators identified by our ligand-receptor analysis (*S100A8*, *S100A9* and *SERPINB3*) (p.10 and Figure 5E). This fibroblast-keratinocyte crosstalk was not fully recapitulated by IL-24 supplementation of the control medium, indicating synergy of IL-24 with other cytokines. We have amended the discussion (p.15) to explicitly state that the identification of these additional mediators will require further in-depth experiments.

3) The authors could do a much better job relating their results to other single cell studies of psoriasis in the literature, including but not limited to PMID: 34279540 (APCs), 33932468 (APCs and keratinocytes), and 35693818 (keratinocytes).

We agree with the reviewer that we should have provided more context for our findings. We have now clarified that in lesional skin, we observed the differential expression of the same keratinocyte genes described in PMID 33932468 and 33479125 (p5-6 and Supplementary Figure 2C). We have also related the gene expression profile observed in myeloid cells with that reported in PMID 34279540 (p.7 and Supplementary Figure 3I). Finally, we have reproduced the upregulation of *CCL19* in fibroblasts described in PMID 33479125 (p.9 and Supplementary Figure 5D).

4) Much emphasis is made by the authors on the observation of molecular changes before clinical improvement. This is not new. Many studies in the past have observed molecular changes (by bulk RNA-seq and flow cytometry) in weeks 0 to 4 after therapy (e.g. PMID: 27667537, 30703387, 35757784, many others). Though 3 days is a bit sooner, the main concept has been known for a while. Please acknowledge the context of these prior studies and reduce claim of novelty.

We now explicitly refer to the context provided by prior bulk RNA-seq studies (p.5), without making claims of novelty.

REVIEWER #2:

The manuscript by Francis L et al presents highly interesting data on early resolution of psoriasis skin lesions following anti-IL13 treatment. Identification of a distinct pro-inflammatory fibroblast cell population limited to active psoriasis lesions which rapidly decreases following treatment start is novel and offers insight into the early initiating molecular events in resolution of psoriasis inflammation. The study involves 5 clinically homogenous individuals biologically naive males with longstanding severe psoriasis.

Skin biopsies, before treatment start, day 3 and day 14 are obtained from site-matched areas- areas however not specified - please provide this information.

We have now clarified (Methods, p.16) that all biopsies were obtained from the lower back/buttocks.

Single cell analysis was performed according to established methodology, epidermal and dermal cells were disassociated and classified in clusters obtaining single cell RNA. The main finding was a psoriasis specific WNT5+/IL24 fibroblast population involved in IL17 signalling detected at Day 0 where ligand receptor interaction analysis pointed towards an interaction with its receptor on spinous keratinocytes. The WNT5+/IL24 fibroblast population declined already at Day 3 and was absent at Day 14 presumably leading to declining IL17 effect on keratinocytes identified as an early and important mechanisms for the resolution of pso inflammation. Validation in public transcriptomic datasets showed similar effects in patients treated with other anti-psoriasis drugs (anti-IL17, and even topical therapy) supporting the idea that this is a general mechanism occurring in resolving pso.

The data are convincing but the study would be much enhanced by adding up-to-date spatial architecture through comprehensive spatial transcriptomics revealing interacting cell clusters in situ.

We have re-analysed a psoriasis spatial transcriptomic dataset (Castillo et al, Sci Immunol 2023, PMID 37267384), comprising 25 samples obtained from healthy, lesional and non-lesional skin. While *IL24* expression was virtually undetectable, we were able to identify our population of interest as *WNT5A+/COMP+* cells. We found that the spots expressing these markers are virtually exclusive to psoriasis lesional skin samples and are present at the dermal-epidermal junction, which is in line with our *in-situ* hybridization findings (p.10 and Supplementary Figure 6, B and C). By comparing gene expression in *WNT5A+/COMP+* vs all other spots we observed an upregulation of spinous/supraspinous keratinocyte markers and proinflammatory genes detected in our ligand-receptor analysis (*S100A8*, *S100A9*, *SERPINB3*) (Supplementary Figure 6D). These findings indicate that *WNT5A+/IL24+* fibroblasts are located in an inflammatory microenvironment, in close proximity to spinous/supraspinous keratinocytes, which are likely to be their main interaction partners.

We further investigated this inflammatory crosstalk by generating an in-vitro model of the *WNT5A+/IL24+* state and demonstrating that the culture medium from these cells upregulated *S100A8*, *S100A9* and *SERPINB3* expression in primary human keratinocytes (p.10 and Figure 5E).

REVIEWER #3:

This is a paper by Francis and colleagues on sc analysis of psoriasis focusing on fibroblasts. The authors obtained skin biopsies from patients with psoriasis treated with anti-IL-23 neutralizing antibodies. The most prominent changes observed on day 3 were observed in myeloid cells and fibroblasts, on day 14 myeloid cells and keratinocytes. They found an overrepresentation of IL-17-related genes in keratinocytes and its downregulation in spinous and supraspinous keratinocytes. When analyzing immune cells, they found that mainly terminally differentiated effector memory T cells responded to IL-23 blockade. On day 14 certain DC3 subsets showed the strongest response to IL-23 blockade. Effects of IL-23 inhibition on vascular endothelial cells and pericytes were also investigated. Finally, the authors focused on the effects of IL-23 blockade on fibroblasts and describe a significant decrease in the frequency of WNT5A+/IL24+ cells. They also studied the possible interactions of such fibroblasts with keratinocyte proliferation and differentiation. The finding of WNT5A+/IL24+ fibroblasts in psoriatic skin and its regulation was validated in three other psoriasis datasets from public repositories (treatment with another IL-23 inhibitor or IL-17 inhibitor or topical steroids).

This is a well-written elegant paper on psoriasis, deciphering the changes in immune and tissue cell subsets during treatment with IL-23 targeting neutralizing antibodies. The focus is on early responses at day 3 and 14.

What is missing in this paper is functional data. What exactly is the role of WNT5A+/IL24+ fibroblasts in psoriasis? What are the signals that induce IL-24 in fibroblasts? IL-23, IL-17 or a combination of cytokines (TNF,...)? Do the WNT5A+/IL24+ fibroblasts express the IL-23 receptor?

To address these important questions, we have generated additional in-silico and experimental data. We now show that the genes that are upregulated by TNF and IL-17 in fibroblasts are enriched among those that are preferentially expressed in WNT5A+/IL24+ cells. Accordingly, we readily detected the expression of the TNF and IL-17 receptors (but not that of IL23R), in WNT5A+/IL24+ cells (p.9 and Supplementary Figure 6A). To experimentally validate these findings, we treated primary human fibroblasts with TNF and IL-17A. We found that the two cytokines synergize to upregulate the expression of marker genes for the WNT5A+/IL24+ cell state (WNT5A, IL24, IL8 and TBX3) (p.9 and Figure 5D).

The authors should add some data from functional assays (e.g., in vitro co-cultures) on WNT5A+/IL24+ fibroblasts and keratinocytes. What is the impact of WNT5A+/IL24+ and IL24- (negative) fibroblast populations on keratinocyte proliferation, differentiation and production of secreted factors?

To build on our ligand-receptor analysis suggesting an interaction of WNT5A+/IL24+ cells with keratinocytes, we cultured IL-17A/TNF stimulated human primary fibroblasts as an in-vitro model of the WNT5A+/IL24+ state. We then transferred supernatants from these cells to primary human keratinocytes. We found that this did not affect the expression of proliferation or keratinocyte differentiation markers (p.10 and Supplementary Figure 6E), but led to the upregulation of pro-inflammatory mediators identified by our in-silico analysis (S100A8, S100A9 and SERPINB3) (p.10 and Figure 5E). This fibroblast-keratinocyte crosstalk was not fully recapitulated by IL-24 supplementation of the control medium, suggesting a role for other cytokines. We have amended the discussion (p.15) to explicitly state that the identification of these additional mediators will require further in-depth experiments.

REVIEWERS' COMMENTS

Reviewer #1 (Remarks to the Author):

The authors have done a nice job responding to the reviewer comments and concerns. As a result, the manuscript is significantly improved.

Reviewer #2 (Remarks to the Author):

The authors have performed extensive additional experiments requested from reviewers and clarified important issues in the text. The resulting functional data and more detailed analysis of cellular localization have much improved the quality. It is a timely and interesting work advancing our understanding of signaling in skin inflammation focusing on the only recently recognized role for supportive cells (ie in this context fibroblasts) in regulating inflammatory signals in psoriasis. I recommend for publication

Reviewer #3 (Remarks to the Author):

This is a revised paper by Francis et al. on inflammatory fibroblasts in psoriasis studied by single-cell analysis. The authors have addressed most of the points raised by the three referees. They added additional data, especially some functional and co-culture data from stimulated fibroblasts and co-cultured keratinocytes and analysis (comparison of single-cell data from atopic dermatitis and prurigo nodularis fibroblasts). Taken together the paper improved. Although there is a need on more investigations of inflammatory fibroblasts in skin diseases like psoriasis, the revised manuscript is now acceptable.